# Near-atomic resolution structures of interdigitated nucleosome fibres

Zenita Adhireksan[1,2,3], Deepti Sharma [1,2,3], Phoi Leng Lee[1,2,3] & Curt A. Davey [1,2✉]

Chromosome structure at the multi-nucleosomal level has remained ambiguous in spite of its central role in epigenetic regulation and genome dynamics. Recent investigations of chromatin architecture portray diverse modes of interaction within and between nucleosome chains, but how this is realized at the atomic level is unclear. Here we present near-atomic resolution crystal structures of nucleosome fibres that assemble from cohesive-ended dinucleosomes with and without linker histone. As opposed to adopting folded helical '30 nm' structures, the fibres instead assume open zigzag conformations that are interdigitated with one another. Zigzag conformations obviate extreme bending of the linker DNA, while linker DNA size (nucleosome repeat length) dictates fibre configuration and thus fibre–fibre packing, which is supported by variable linker histone binding. This suggests that nucleosome chains have a predisposition to interdigitate with specific characteristics under condensing conditions, which rationalizes observations of local chromosome architecture and the general heterogeneity of chromatin structure.

[1] School of Biological Sciences, Nanyang Technological University, 60 Nanyang Drive, Singapore 637551, Singapore. [2] NTU Institute of Structural Biology, Nanyang Technological University, 59 Nanyang Drive, Singapore 636921, Singapore. [3]These authors contributed equally: Zenita Adhireksan, Deepti Sharma, Phoi Leng Lee. ✉email: davey@ntu.edu.sg

Eukaryotic DNA is packaged by basic proteins into chromatin, which consists of repeating nucleosome units composed of a core element of ~146 base pairs (bp) wrapped around an octamer of four different core histone proteins in $1\frac{2}{3}$ left-handed turns[1]. The nucleosome core elements are connected by a variable length of typically 10–90 bp of linker DNA, yielding the nucleosome, which can be associated with a 5th type of histone protein, linker histone[2]. The structural and chemical disposition of chromatin components beyond the DNA sequence, epigenetic features, influence DNA accessibility and nuclear factor activity, which enable control of genomic activities in the cell with temporal and spatial precision.

Condensing of nucleosome chains into compact states is modulated by several distinct types of architectural factors, including linker histones, which are essential proteins in higher eukaryotes that serve critical functions in gene regulation, maintaining genomic integrity and generating mitotic chromosomes for cell division[3,4]. We have an extensive understanding of how epigenetic features of individual nucleosomes—including nucleosome structure, stability, dynamics and positioning as well as DNA methylation, core histone post-translational modifications and variant localization—influence genome activities. However, our comprehension of how nucleosomes are organized into higher order structures is comparatively lacking, and yet this is essential for understanding genomic function. The reason why our grasp of chromatin higher order organization is relatively poor compared to the structure and function of individual nucleosomes is twofold in that in vitro characterization involves studying very large assemblies that are intrinsically dynamic in nature. Second, at the cellular level, imaging techniques that afford the capacity to resolve individual nucleosomes in situ are just coming of age[5].

Up until the last decade or so, the textbook consensus on chromatin structure was that the subsequent compaction level of the nucleosome fibre took the form of folded, helical, so-called 30 nm, conformations, for which there are two general models, either a solenoidal or a two-start zigzag configuration[6,7]. Indeed, 30 nm fibres form in vitro under carefully controlled conditions at low divalent metal cation concentrations with regularly spaced nucleosome arrays[8–10] and with chromatin isolated from various cell types[11–14]. However, in recent years, a number of studies applying a variety of imaging and analytical methods to in situ interphase and mitotic chromatin as well as that isolated from cells have provided evidence of compact chromatin consisting of irregular interdigitated fibres with zigzag characteristics, but not 30 nm structures[5,15–24]. Moreover, under condensing $Mg^{2+}$ concentrations, nucleosome arrays have been shown to form large globular structures that consist of extended nucleosome fibres, which are interdigitated with one another[10]. Nevertheless, 30 nm fibre conformations have been reported to occur in the context of terminal differentiation[25–28], consistent with special structural features associated with transcriptionally silenced chromatin[29]. Therefore, the key questions remain as to what factors determine whether chromatin assumes interdigitated or 30 nm configurations and if the former coincides with a common mode of organization in vivo, then how is this architecture realized at the atomic level.

As a solid-state method, pioneering crystallographic investigations have yielded detailed insight into chromatin compaction and higher order structure[30–35]. However, the approach is challenged by the dynamic nature of nucleosomal systems, limiting the ability to obtain well-ordered crystals of large assemblies. Indeed, few crystal structures of nucleosomes (i.e., >147 bp) have been obtained, and those reported are based on blunt-ended DNA fragments that coincide with a lack of double helix continuity in the lattice and typically significant disorder that limits

resolution. Here, we present findings from cohesive-ended dinucleosomes that self-assemble into lattices composed of fibres with uninterrupted Watson–Crick continuity of the double helix. The interdigitated nucleosome fibre structures shed light on chromosome architecture and help rationalize seemingly conflicting reports on chromatin structure.

## Results

**Dinucleosomes assemble into open zigzag fibres.** We obtained nucleosome fibre crystals with two different dinucleosome constructs of 349 and 355 bp that differ with respect to the lengths of the linker DNA (Figs. 1 and 2; Supplementary Fig. 1; Supplementary Movies 1 and 2). The 145 bp nucleosome core elements of both constructs consist of a near maximal histone octamer-affinity, Widom-based sequence[36–38] flanked by histone octamer-refractory poly-A|T elements in the linker DNA arms, followed by mixed sequence elements that terminate in single-stranded 3′ TGCA overhangs. In divalent metal-containing buffers, the dinucleosomes assemble into a lattice in which the four-nucleotide cohesive termini associate through Watson–Crick base pairing (Figs. 1 and 2; Supplementary Fig. 2). The offset pairing of dinucleosomes in an open-ended fashion yields uninterrupted continuity of the DNA double helix from one end of the crystal to the other, making the lattice like a fabric woven of chromatin fibres that are interdigitated with one another. The length of the fibres composing the lattice are estimated to be up to roughly $10^5$ nucleosomes or 18 million bp.

Nucleosome fibre structures for the 349 and 355 constructs were solved at up to 3.4 Å and 3.8 Å resolution, respectively, and in both cases the asymmetric unit is a single dinucleosome repeat (Figs. 1 and 2; Tables 1–3; Supplementary Fig. 1; Supplementary Movies 1 and 2). Fibres of the 355 construct are highly extended, having an average rise of 42.5 Å per nucleosome, with the dinucleosome repeats situated perpendicular to the fibre axis and stacked edgewise one on top of the other to create a pseudo-twofold symmetric double-sided staircase. The 31 bp 'shared' linker DNA sections that connect the two nucleosomes within a dinucleosome cross over roughly orthogonal to the fibre axis, whereas the 34 bp 'paired' (annealed) linker DNA sections that connect adjacent dinucleosomes are aligned more coincident with the fibre axis. As such, the fibre has a decisive zigzag configuration comprising two parallel nucleosome columns, where each nucleosome connects to two others in the opposing column. Due to the extended nature of the fibre, the stacking interactions between nucleosomes in a given column are sparse, but entail both divalent metal assisted DNA-DNA interdigitation and histone-DNA contacts involving symmetry-related elements at the nucleosome core periphery (Fig. 3).

In spite of differing by 2 and 4 fewer bp, respectively, in the shared and paired linker DNA sections, the 349 nucleosome fibre is analogous to that of the 355 in that it has a zigzag configuration comprising two parallel nucleosome columns (Figs. 1 and 2). However, the two fibre types are otherwise remarkably distinct. Instead of the fibre axis running between the two nucleosomes of the dinucleosome repeats as occurs for 355, the 349 fibre axis runs between the repeats, which stack with one another in an offset fashion within each column. In this regard, the nearly planar 349 dinucleosomes are canted along the fibre axis at a roughly 45° angle, supporting an alternating arrangement where each dinucleosome is paired in turn with two different dinucleosomes of the opposing column. Moreover, one face of each nucleosome is stacked against the face of a neighbouring nucleosome with which it is connected DNA-wise via an intervening dinucleosome from the other column. In this way, the 349 fibre is in essence somewhat folded and is thus less extended than that of the 355,

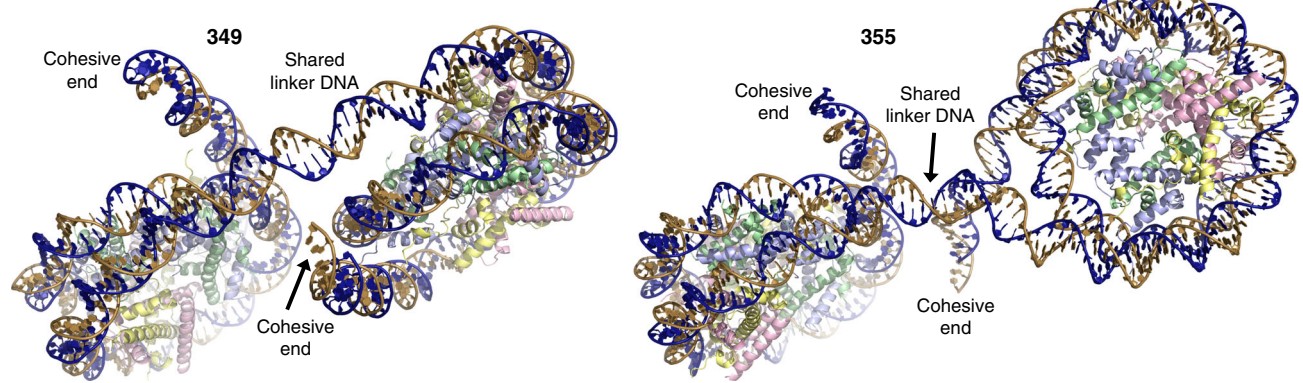

**Fig. 1 Dinucleosome constructs that assemble into fibres.** DNA sequences (single strand, 5'–3' direction) and structures for the 349 and 355 dinucleosomes are shown. Above, the base pairs at the nucleosome centres (dyads) are highlighted in bold, base pair insertions of 355 relative to 349 are underlined and the bars designate the midpoints of pairing of the GTAC cohesive ends between dinucleosomes in the fibre. Below, the dinucleosome structures correspond to the repeating (asymmetric) units in the fibres. The two DNA strands are coloured blue and gold, and the core histone proteins are pale yellow (H2A), pink (H2B), purple (H3) and green (H4).

with an average rise of 30.3 Å per nucleosome. In fact, the orientation of the 30 bp paired linker DNA segments is such that they run substantially retrograde (i.e., negative rise) to the overall translational sense of the fibre. The intrafibre nucleosome face-to-face contacts of the 349 configuration involve histone-histone and histone-DNA contacts as well as divalent metal-mediated interactions (Fig. 3).

**Linker DNA dictates fibre structure**. Given that the nucleosome core regions have identical DNA sequences and conserved structures, the pronounced differences between the 349 and 355 fibre configurations illustrate how the linker DNA dictates fibre arrangement. The influence of the linker DNA structure can be understood in two separate respects—DNA length/twist relationship as opposed to bending of the double helix. Since the two constructs have identical DNA sequences with the exception of the, relative to 349, introduction of 2/4 additional bp of mixed sequence to the shared/paired linker arms of 355 (Fig. 1), differences in bending are a result of context dependency, as opposed to sequence-specific effects. On the other hand, the bp insertions of 355 coincide with double helix twist extensions, relative to the 349 fibre, of 54° (shared) and 88° (paired) between the respectively linked nucleosome core regions. As such, the angular disposition between linked nucleosomes is highly distinct for the two fibre structures (Figs. 1 and 2; Supplementary Movies 1 and 2). Nucleosomes connected by the shared linker arms are nearly co-planar in 349 as opposed to being oblique in 355. The ~90° difference in the paired linker arms means that, while the 349 shared linker DNA sections criss-cross one another in very close proximity (<8 Å; separated by 2 intervening

nucleosomes) at right angles, those of the 355 are widely separated and run parallel to one another. Therefore, the twist differences imposed by differing linker DNA lengths alone have a dominating effect on fibre configuration.

In addition to dramatic rotational distinctions brought about by the small changes in linker DNA length between the 349 and 355 fibres, there are also significant differences in bending of the linker DNA (Figs. 1 and 2). However, as outlined above, these structural differences appear to result from the interplay of orientational constraints imposed by linker DNA length and the optimization of nucleosome packing interactions. In this regard, linker DNA bending deformations are more pronounced in the 355 compared to the 349 fibre, and in fact each linker section has 5–7 flexible (pyrimidine-purine) bp steps between the flanking poly-A|T tracts. In the 349 fibre, the shared linker section displays a gentle arcing, with a 43° bend between the entry/exit points of two linked nucleosome core regions, while the paired linker section is nearly straight (19° bend). The corresponding shared and paired linker arms in the 355 fibre show an overall bending of 26° and 60°, respectively, but both arms also display some excess curvature as the double helix wriggles. By allowing interfibre groove-groove interdigitation, this translational writhing in the 355 linker DNA facilitates close packing of the fibres (see below). Moreover, the pronounced curvature in the 355 paired linker arm may help accommodate the greater difference between its shared and paired linkers (31 versus 34 bp) compared to those in the 349 (29 versus 30 bp).

**Fibres integrate and interdigitate**. In both the 349 and 355 systems, the fibres making up the lattice run parallel to and are

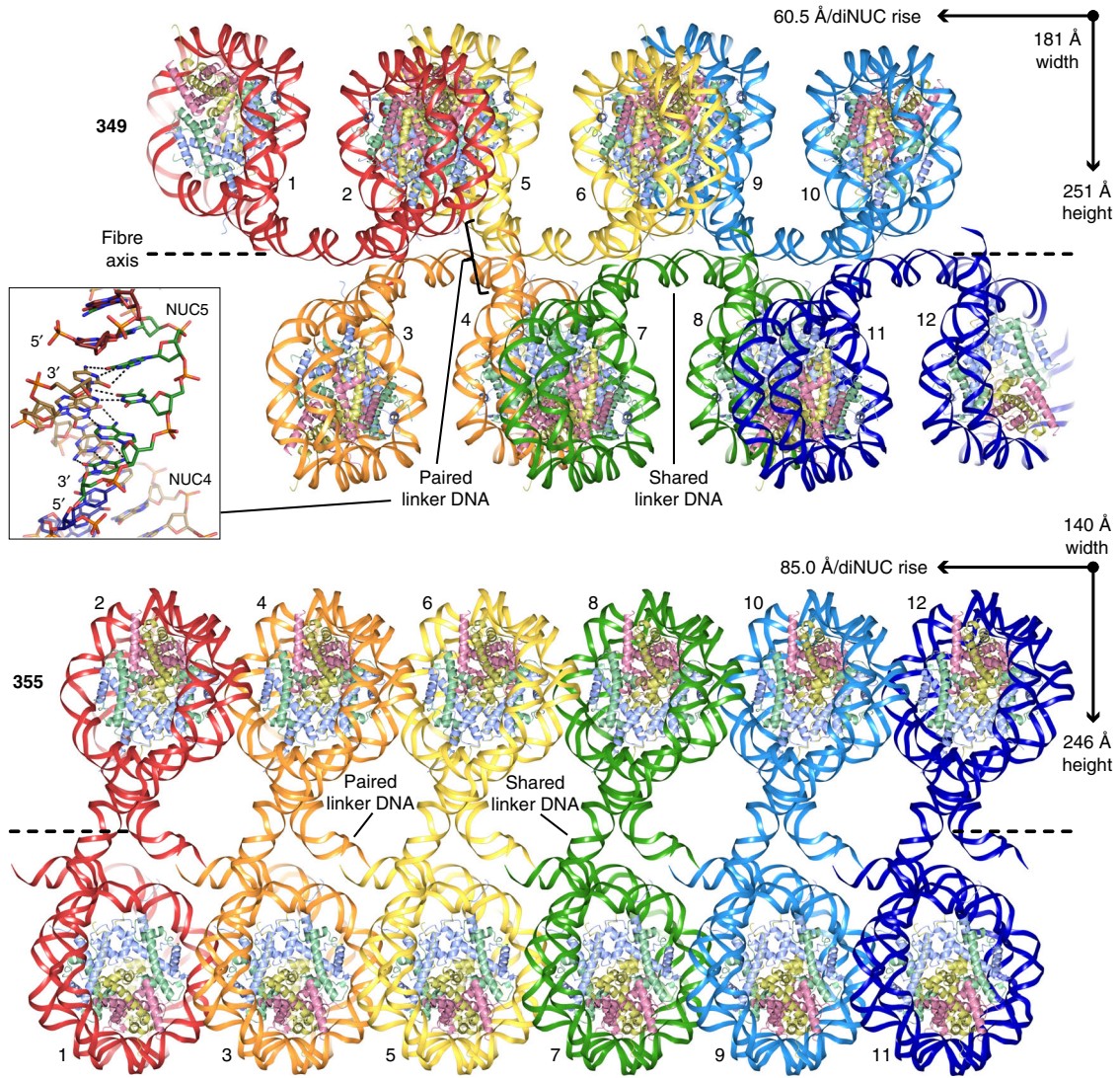

**Fig. 2 Structures of nucleosome fibres.** 12-nucleosome sections of the 349 and 355 fibres are shown. The DNA is coloured to distinguish the dinucleosome repeats. The cores histones are shown in pale yellow (H2A), pink (H2B), purple (H3) and green (H4). The inset depicts the Watson–Crick base pairing from annealing of the cohesive ends between dinucleosomes, generating the paired linker DNA sections.

interdigitated with one another (Fig. 4; Supplementary Movies 3 and 4). Several features are common to both systems, including offset overlapping of the fibres, such that only one nucleosome column of a given fibre stacks against that of a neighbouring fibre. The other is that the columns of both fibre types display a staircase-like profile that permits a key in lock style of fibre–fibre integration. As such, the similar packing configurations between fibres comprise nucleosomal face-to-edge contacts whereby the interacting nucleosomes have roughly perpendicular orientations with respect to one another (Fig. 5a). These inter-fibre interfaces involve histone-DNA contacts as well as divalent metal-mediated interactions (Fig. 5b).

Aside from the interdigitation features common to both the 349 and 355 fibres, one notable distinction relates to the inter-fibre register of the nucleosome columns (Fig. 4; Supplementary Movies 3 and 4). For the 349, the nucleosome cores are juxtaposed with one another such that the linker DNA sections comprise an isolated zone between the two nucleosome columns within a fibre. On the other hand, the 355 nucleosome cores are offset with one another, so that there are contacts between the nucleosome cores of one fibre and the linker DNA of neighbouring fibres. As mentioned above, this coincides with

distinct DNA bending conformations assumed by the 355 linker arms.

**Linker histone mediates intra- and inter-fibre contacts.** We grew crystals of 349 and 355 assembled without linker histone and with saturating amounts of the H1.0 linker histone variant (Tables 1–3; Supplementary Figs. 3 and 4). In either case, fibre structure and packing are nearly identical (r.m.s.d. of super-imposed dinucleosomes assembled with and without linker histone is 0.62 and 0.24 Å for the 349 and 355 systems, respectively), suggesting that a dominant role is played by the nucleosome packing and nucleosome-nucleosome interaction forces. For the 355 system, linker histone binding coincides with either dynamic and/or static (mixed binding modes) disorder, such that there is insufficient clarity of the electron density to model the linker histones. In the 349 crystals, we observe two distinct linker his-tone binding modes, one of which corresponds to the on-dyad mode observed in several earlier structures of nucleosome-linker histone assemblies[31,32,34] (Fig. 6; Supplementary Movie 5). For this binding configuration, the linker histone globular domain sits at the nucleosome centre (dyad), where it interacts extensively

**Table 1 Data collection and refinement statistics for 349 dinucleosome assembled with linker histone (H1.0) and crystals stabilized in either a low (L) or high concentration (H) of cryoprotectant.**

|  | 349-H1.0-L | 349-H1.0-H |
|---|---|---|
| Data collection |  |  |
| Space group | $P2_12_12_1$ | $P2_12_12_1$ |
| Cell dimensions |  |  |
| $a, b, c$ (Å) | 121.06, 184.07, 223.37 | 120.78, 172.88, 216.04 |
| $\alpha, \beta, \gamma$ (°) | 90, 90, 90 | 90, 90, 90 |
| Resolution (Å) | 3.40–49.32 (3.40–3.59)[a] | 3.70–99.01 (3.70–3.90)[a] |
| $R_{merge}$ (%) | 15.1 (248) | 21.2 (164) |
| $R_{pim}$ (%) | 4.4 (72.7) | 7.0 (57.5) |
| $I/\sigma I$ | 12.1 (1.1) | 7.8 (1.4) |
| CC½ (%) | 100 (41.0) | 99.8 (30.9) |
| Completeness (%) | 99.8 (98.6) | 97.6 (94.2) |
| Redundancy | 13.3 (13.1) | 10.6 (9.0) |
| Refinement |  |  |
| Resolution (Å) | 3.40–49.32 | 3.70–91.78 |
| No. reflections | 67,687 | 46,780 |
| $R_{work}/R_{free}$ (%) | 20.6/26.5 | 19.4/26.6 |
| No. atoms | 27,058 | 27,560 |
| Core histone | 12,149 | 12,075 |
| Linker histone | 568 | 1,128 |
| DNA | 14,311 | 14,311 |
| Solvent | 30 | 46 |
| $B$-factors (Å²) | 161 | 141 |
| Core histone | 131 | 112 |
| Linker histone | 198 | 210 |
| DNA | 184 | 161 |
| Solvent | 154 | 118 |
| R.m.s. deviations |  |  |
| Bond lengths (Å) | 0.006 | 0.006 |
| Bond angles (°) | 1.26 | 1.44 |

[a]Single crystal data sets; Values in parentheses are for the highest resolution shell.

**Table 3 Data collection and refinement statistics for 349 dinucleosome and 355 dinucleosome assembled with linker histone (H1.0), with crystals stabilized in a low (L) concentration of cryoprotectant.**

|  | 349-L | 355-H1.0-L |
|---|---|---|
| Data collection |  |  |
| Space group | $P2_12_12_1$ | $P2_1$ |
| Cell dimensions |  |  |
| $a, b, c$ (Å) | 120.81, 182.83, 219.03 | 89.13, 229.61, 119.05 |
| $\alpha, \beta, \gamma$ (°) | 90, 90, 90 | 90, 92.71, 90 |
| Resolution (Å) | 5.10–49.11 (5.10–5.59)[a] | 4.90–229.6 (4.90–5.29)[a] |
| $R_{merge}$ (%) | 25.3 (149) | 9.4 (184) |
| $R_{pim}$ (%) | 10.5 (61.7) | 4.0 (83.0) |
| $I/\sigma I$ | 6.3 (1.4) | 9.3 (1.0) |
| CC½ (%) | 99.7 (49.4) | 99.5 (33.1) |
| Completeness (%) | 99.8 (100) | 99.5 (97.7) |
| Redundancy | 6.7 (6.7) | 6.6 (5.7) |
| Refinement |  |  |
| Resolution (Å) | 5.10–49.11 | 4.90–118.9 |
| No. reflections | 19,927 | 21,435 |
| $R_{work}/R_{free}$ (%) | 22.2/28.6 | 19.5/24.6 |
| No. atoms | 26,460 | 26,468 |
| Core histone | 12,149 | 11,911 |
| Linker histone | – | – |
| DNA | 14,311 | 14,557 |
| Solvent | – | – |
| $B$-factors (Å²) | 196 | 291 |
| Core histone | 145 | 232 |
| Linker histone | – | – |
| DNA | 239 | 339 |
| Solvent | – | – |
| R.m.s. deviations |  |  |
| Bond lengths (Å) | 0.005 | 0.005 |
| Bond angles (°) | 1.39 | 1.27 |

[a]Single crystal data sets; Values in parentheses are for the highest resolution shell.

**Table 2 Data collection and refinement statistics for 355 dinucleosome crystals stabilized in either a low (L) or high concentration (H) of cryoprotectant.**

|  | 355-L | 355-H |
|---|---|---|
| Data collection |  |  |
| Space group | $P2_1$ | $P2_1$ |
| Cell dimensions |  |  |
| $a, b, c$ (Å) | 88.99, 228.97, 118.84 | 85.04, 218.60, 115.93 |
| $\alpha, \beta, \gamma$ (°) | 90, 92.68, 90 | 90, 90.22, 90 |
| Resolution (Å) | 4.6–49.28 (4.6–4.92)[a] | 3.81–47.98 (3.81–3.96)[a] |
| $R_{merge}$ (%) | 12.9 (196) | 9.6 (145) |
| $R_{pim}$ (%) | 5.4 (81.8) | 4.0 (64.6) |
| $I/\sigma I$ | 6.6 (1.2) | 11.4 (1.2) |
| CC½ (%) | 99.5 (48.1) | 99.9 (42.4) |
| Completeness (%) | 99.9 (100) | 99.8 (98.7) |
| Redundancy | 6.9 (6.6) | 6.8 (5.8) |
| Refinement |  |  |
| Resolution (Å) | 4.6–49.28 | 3.81–47.98 |
| No. reflections | 25,862 | 40,456 |
| $R_{work}/R_{free}$ (%) | 20.5/26.1 | 20.8/28.6 |
| No. atoms | 26,471 | 26,496 |
| Core histone | 11,911 | 11,934 |
| Linker histone | – | – |
| DNA | 14,557 | 14,557 |
| Solvent | 3 | 5 |
| $B$-factors (Å²) | 269 | 202 |
| Core histone | 211 | 161 |
| Linker histone | – | – |
| DNA | 317 | 236 |
| Solvent | 122 | 147 |
| R.m.s. deviations |  |  |
| Bond lengths (Å) | 0.005 | 0.004 |
| Bond angles (°) | 1.28 | 1.27 |

[a]Single crystal data sets; Values in parentheses are for the highest resolution shell.

additional contacts with this element of the juxtaposed dinucleosome. As such, the on-dyad linker histone mediates internucleosomal, albeit intrafibre, interactions.

In contrast to the on-dyad association, the other linker histone binding site is remote from any nucleosome dyad region and does not involve interactions with linker DNA (Fig. 6; Supplementary Movies 5 and 6). In this 'non-dyad' mode, the linker histone globular domain instead sits in a niche surrounded by nucleosomes, where it interacts with nucleosome core DNA elements from three separate fibres. In this regard, the non-dyad binding linker histone mediates interfibre contacts and thereby helps consolidate the fabric of the interdigitation network.

In comparing the two different linker histone binding modes observed in the 349 fibres, it is notable that in both cases very similar DNA binding motifs are utilized by H1.0 (Fig. 6a, b). This is consistent with the positioning of the double helix elements provided by the nucleosomes from the three fibres in the non-dyad mode mimicking closely the configuration of the two linker DNA plus dyad double helix components coinciding with on-dyad binding. In fact, the nucleosome core regions from the fibres that surround the linker histone in the non-dyad mode could enhance the interaction potential by providing additional DNA binding surfaces.

## Discussion

Here, we have shown that two distinct cohesive-ended dinucleosomes both self-assemble into continuous nucleosome chains under compacting conditions. The chains assume open configurations, in which the DNA path zigzags to form two nucleosome columns on either side of the fibre axis, permitting dense packing of nucleosomes by virtue of the staircase-like fibre profiles that foster a key in lock style of interdigitation. The structures afford a near-atomic resolution picture of how chromatin fibres can adopt

with the outward facing minor groove in addition to making contacts with both linker DNA arms. Given the close proximity of the shared linker DNA arm in the opposing nucleosome column of the fibre, the linker histone globular domain in fact makes

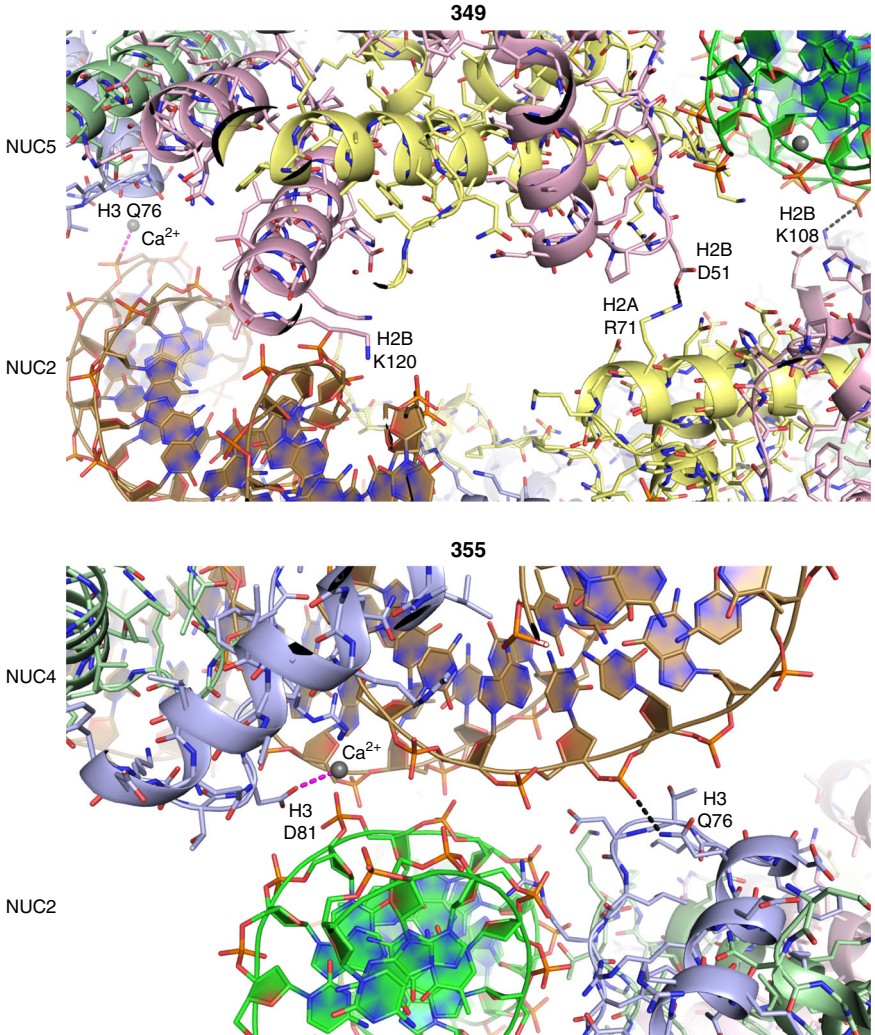

**Fig. 3 Intrafibre nucleosome-nucleosome interactions.** Nucleosome-nucleosome interfaces within the 349 and 355 fibres are shown with dashed lines indicating hydrogen bonds (black) and coordinate bonds (magenta; $Ca^{2+}$ ions, grey spheres). The cores histones are shown in pale yellow (H2A), pink (H2B), purple (H3) and green (H4).

zigzag configurations that interdigitate with one another to produce highly condensed states. This is consistent in an overall sense with investigations of heterochromatin and metaphase chromosome architecture, which show chromatin chain interdigitation[5,15–18,23] with zigzag characteristics of the individual fibres[19,21,22,24]. Furthermore, the chromosome-like, globular nucleosome array condensates that form under compacting divalent metal cation concentrations, similar to the ionic conditions used to obtain and stabilize the present fibre crystals, are also composed of interdigitated chains that would appear to be zigzag[7,10]. From the structures here, the global preference for zigzag fibre conformations is explained by obviating, or at least minimizing, the need for extreme bending of the linker DNA, as such arrangements allow for the nucleosome core entry/exit points to roughly face one another.

While the two fibre structures, 349 and 355, have similar overall features in terms of their zigzag nature and mode of interdigitation, at the local level they are highly distinct. This is a consequence of differences in the lengths of the linker DNA that connect the nucleosome core regions. Although these differences correspond to only several bp per linker arm, by altering the twist of the double helix at the radian level, they have a profound

influence in particular on the orientational relationships between nucleosomes within a fibre and consequently between fibres as well. In this respect, there is an obvious interplay between the fibre configuration adopted and optimization of fibre–fibre interactions (packing), especially given their open, unfolded (355) or only partially folded (349), conformations.

By extrapolation from what we have observed here with two different nucleosomal constructs, there may be many fibre configurations associated with distinct linker DNA lengths. Linker DNA length varies depending on species, cell type, cell status and genomic location, displaying a non-uniform distribution between about 10 and 90 bp[3,39]. This corresponds to a broad distribution of nucleosome size, or nucleosome repeat length, ranging over roughly 160–240 bp. With consideration of only variations in linker DNA length alone, this would suggest on average a prodigious variability in local chain conformation from one section of nucleosomes to the next. We envisage that nucleosome chains composed of mixed linker DNA lengths would interdigitate with each other through maximization of stacking interactions by exploiting the inherent flexibility of open (non-folded) fibres. With further consideration of variations in nucleosome core composition and the presence of other nuclear factors, this could

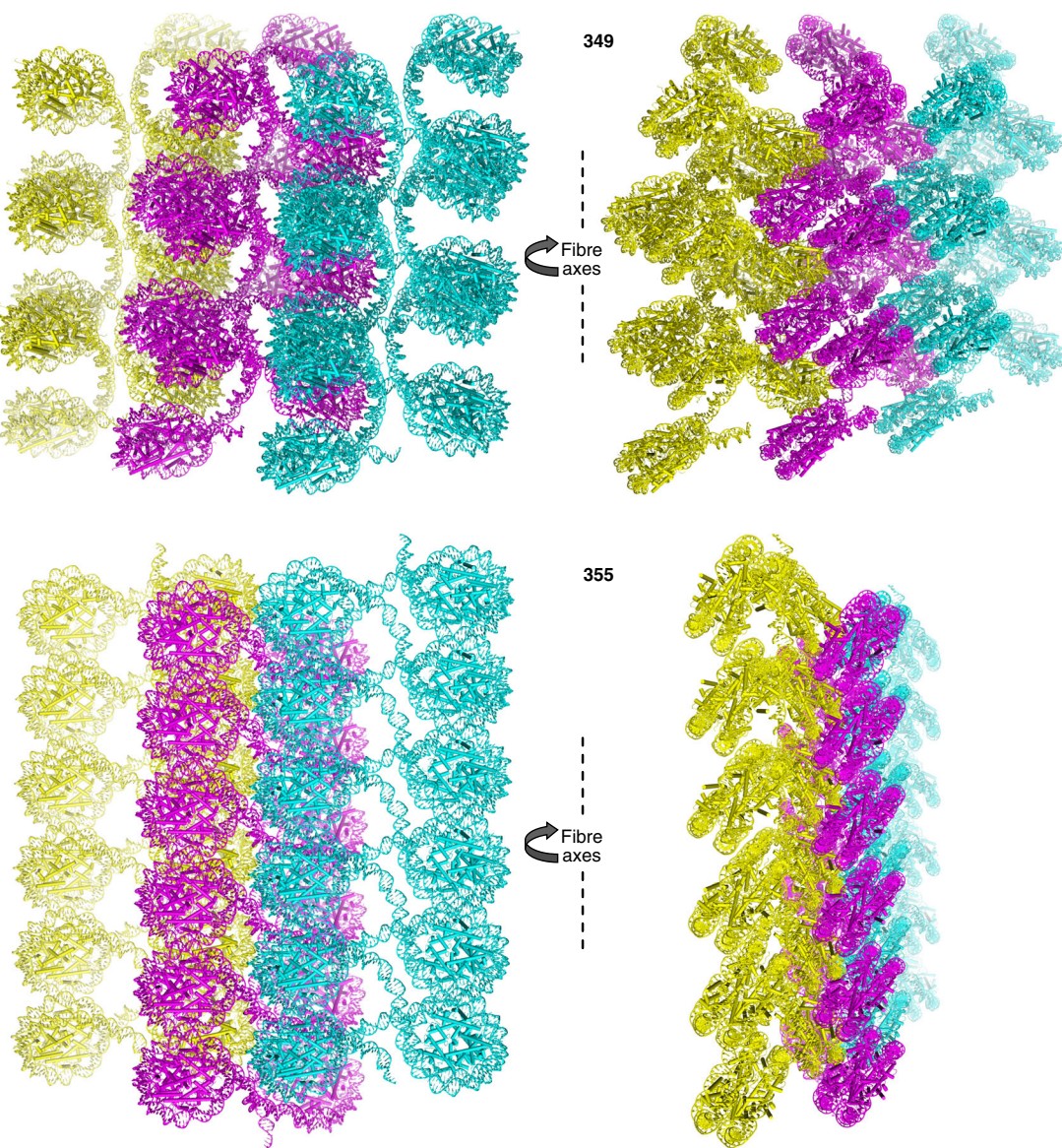

**Fig. 4 Interdigitation of nucleosome fibres.** 12-nucleosome sections for each of three integrated fibres are shown from two perspectives for the 349 and 355 systems. The central fibre is coloured magenta, with the two adjacent fibres in yellow and cyan.

account for the observed heterogeneity of chromatin structure at the multi-nucleosomal level[5,20]. Indeed, we find contacts of all sorts that stabilize nucleosome-nucleosome interactions in the fibre structures, which include the core histones, linker histones, DNA and divalent metal ions. While the highly basic N-terminal tail domains of the core histones are for the most part too disordered to model in the fibres, they have been seen to mediate a variety of internucleosomal contacts in crystal structures of nucleosome core particles[40]. Therefore, the histone tails are likely to mediate potentially numerous interactions with neighbouring nucleosomes within and between fibres, as we observe for instance with H4 R23 in the 349 structure (Fig. 5b). Moreover, non-tail core histone elements, in particular the protruding C-terminal α-helix extension of H2B, stabilize specific intra- and inter-fibre interactions (Figs. 3 and 5b), which could be impacted by post-translational modifications (e.g., H2B K120 acetylation/ ubiquitination). As such, in vivo, variations in histone variant composition, post-translational modifications and the presence of architectural factors, in addition to the activity of other cationic species, would be expected to have a significant impact on fibre

structure or dynamics in a localized fashion, as has been suggested by a multitude of chromatin investigations.

Our results support the premise that chromatin chains have a predisposition to interdigitate by adopting zigzag conformations, and this is consistent with the present general consensus on cellular nucleosome organization[3,7]. Furthermore, with the system studied here, we find that this tendency of nucleosome chains to interdigitate is unaltered by the presence or absence of linker histone, which is observed to support interdigitation configurations through binding in variable modes. Although it remains to be established what linker histone binding modes are prevalent in vivo, the findings here suggest that these architectural factors could bind opportunistically in structural niches to stabilize compact chromatin. Nonetheless, there are conflicting reports in the literature regarding whether or not chromatin can assume folded helical 30 nm structures in vivo. Some experts have suggested that 30 nm fibre structures may exist only transiently or in the context of specialized regulatory features. This would be consistent with reports of 30 nm structures coinciding with states of terminal differentiation[25–28]. Moreover, our findings here

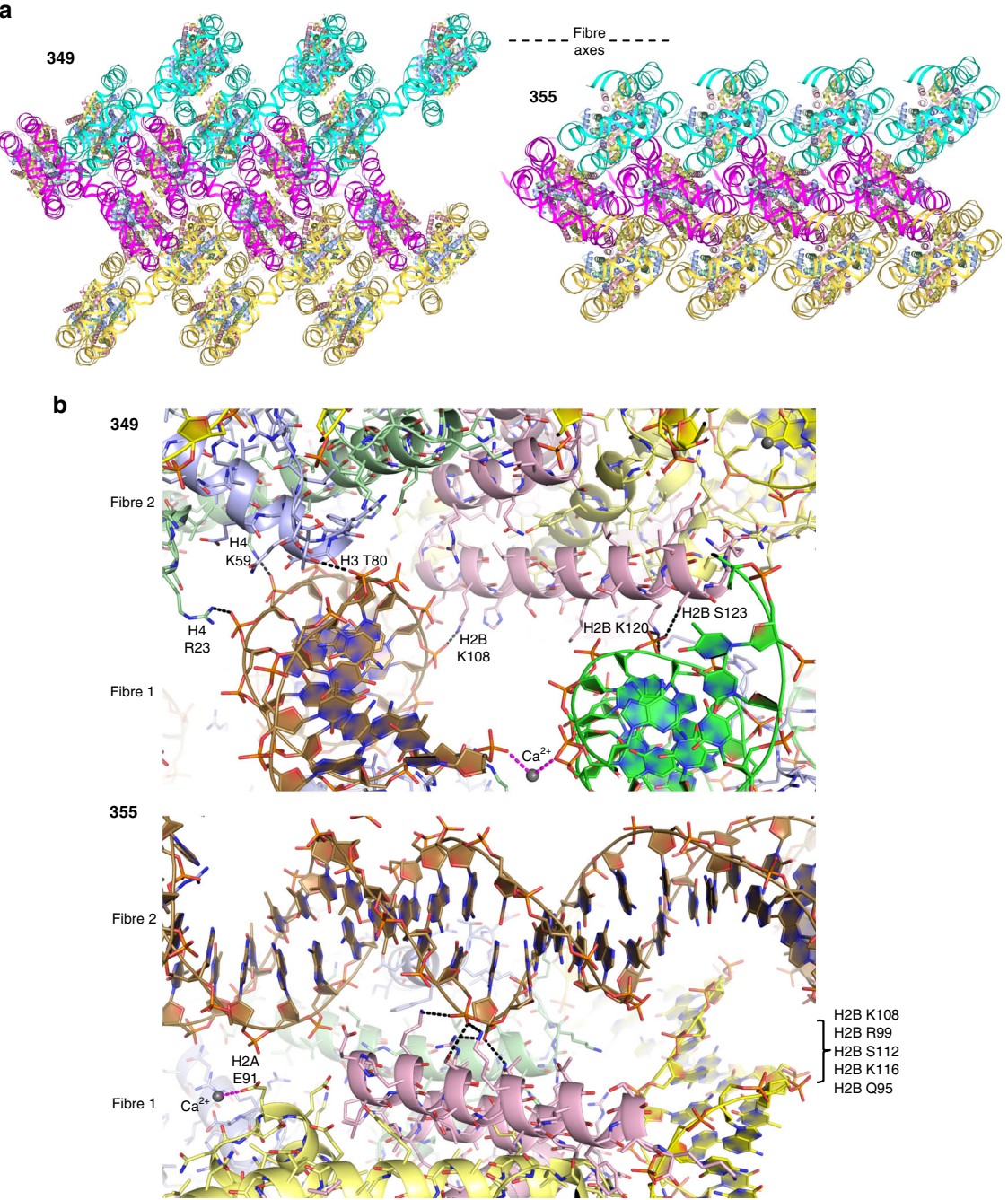

**Fig. 5 Interdigitation interfaces between nucleosome fibres. a** A single nucleosome cross-section shows the nucleosomal edge-to-face stacking interactions that support the interdigitation of fibres. 6-nucleosome (349) and 4-nucleosome (355) sections are shown for each of three fibres, wherein the DNA is coloured cyan, magenta or yellow. **b** Interfibre nucleosome-nucleosome interfaces are shown with dashed lines indicating hydrogen bonds (black) and coordinate bonds (magenta; $Ca^{2+}$ ions, grey spheres). **a**, **b** The cores histones are shown in pale yellow (H2A), pink (H2B), purple (H3) and green (H4).

emphasize the critical role played by the length of the linker DNA, and consequently its regularity from nucleosome to nucleosome. Studies on 30 nm structures have shown that folding is also dependent on linker DNA size[9,41,42], and this is consistent with reports of 30 nm fibre structures present in centromeric regions where nucleosomes are positioned in well phased regular arrays[43]. Alternatively, the presence or action of other nuclear factors—not just histone and architectural proteins but possibly also chromatin remodelling activities—may be involved in promoting 30 nm fibre folding.

In recent times, there have been a number of breakthrough investigations shedding light on nucleosome organization in vivo. Some of these works utilize direct imaging methods while others rely on sophisticated proximity mapping strategies that yield internucleosomal contact relationships on the order of 100s of bp in separation[19,24,29]. The 349 and 355 fibre models are in remarkable agreement with these proximity profiles in showing enrichment of N to N + 2 contacts (interactions between nucleosomes N and N + 2; 349 and 355) but also of N to N + 3 contacts (349 only), albeit to a lesser extent (Fig. 7). Importantly,

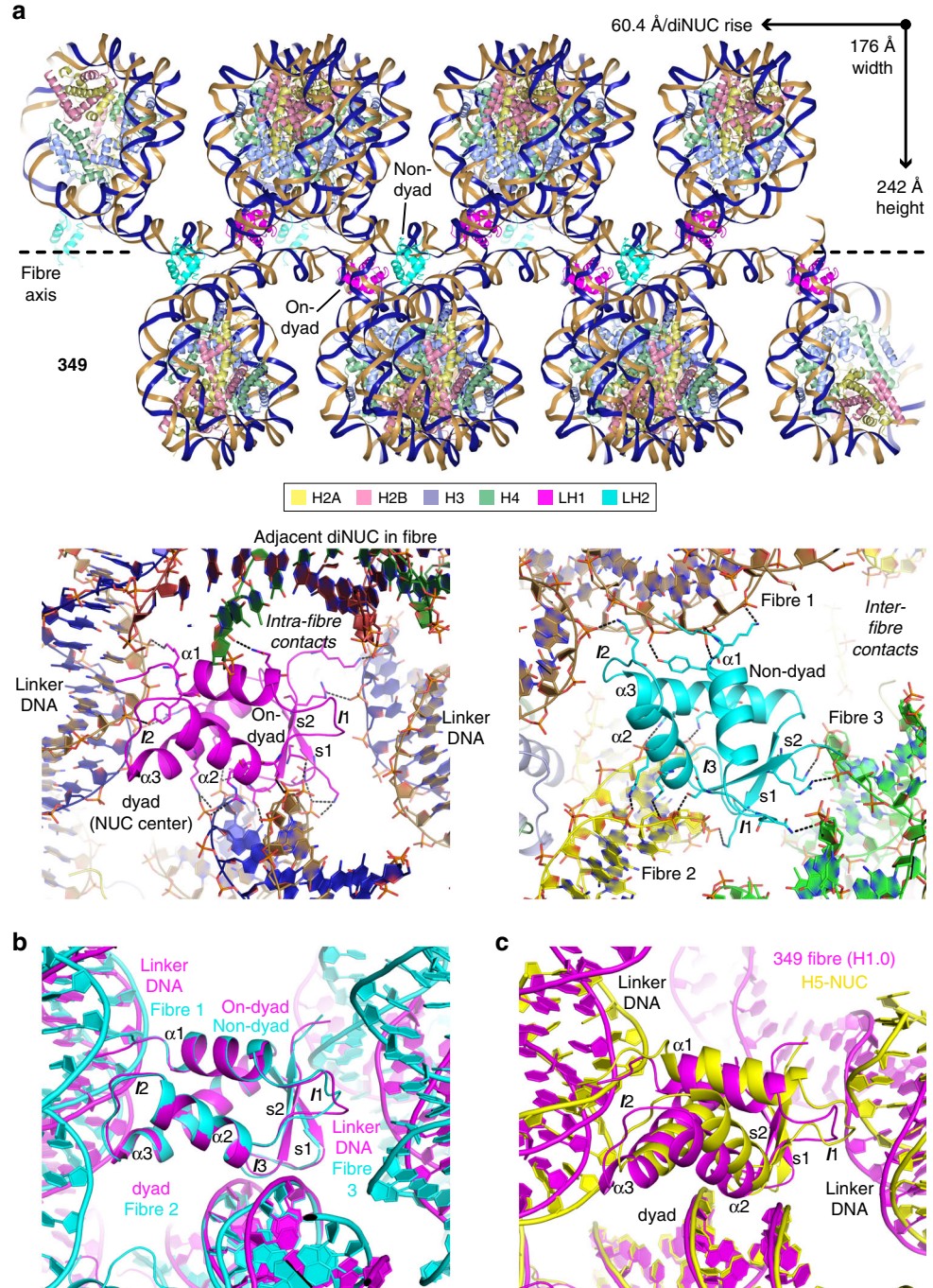

**Fig. 6 Linker histone binding to nucleosome fibres. a** A 12-nucleosome section of the 349 fibre is shown at top, with the DNA coloured as two strands to illustrate the continuity of the double helix. Close views of linker histone association are shown below, with dashed black lines indicating hydrogen bonds. Linker histones are coloured to distinguish the on-dyad (magenta; LH1) and non-dyad (cyan; LH2) binding modes. **b** Similarities between the on-dyad and non-dyad linker histone binding modes in the 349 structure. The two linker histone structures were least-squares superimposed and are shown together with their corresponding interaction environments in the superimposed frame. **c** Comparison of the 349 on-dyad linker histone association (magenta) with that of the H5 globular domain bound to a 167 bp nucleosome[31] (yellow). The two linker histone-nucleosome structures were least-squares superimposed. Differential positioning of the linker DNA arms coincides with a corresponding shift in localization of the linker histone globular domain. **a, b, c** Secondary structural elements of the linker histone globular domain are indicated (α alpha-helices; s beta-strands; l loops).

the fibre models fit the 'α-structure' (tetrahedron) and 'β-structure' (rhombus) chromatin folding motifs determined for yeast[24] and provide a means to fit/interpret a proximity map of human cellular chromatin[29] without the need for invoking folded helical 30 nm structures. Nevertheless, the human proximity map revealed exciting distinctions in structure between open (non-

compact), H3K9me3-marked silenced and H3K27me3-marked silenced chromatin. In fact, the 355 fibre structure is remarkably similar to the 2-start zigzag form of 30 nm structure[42]. Both share dominant N + 2 contact relationships from the zigzag arrangement of the chain, but the 355 configuration (246 Å height, 140 Å width and 552 Å length over 12 nucleosomes) is as if one had

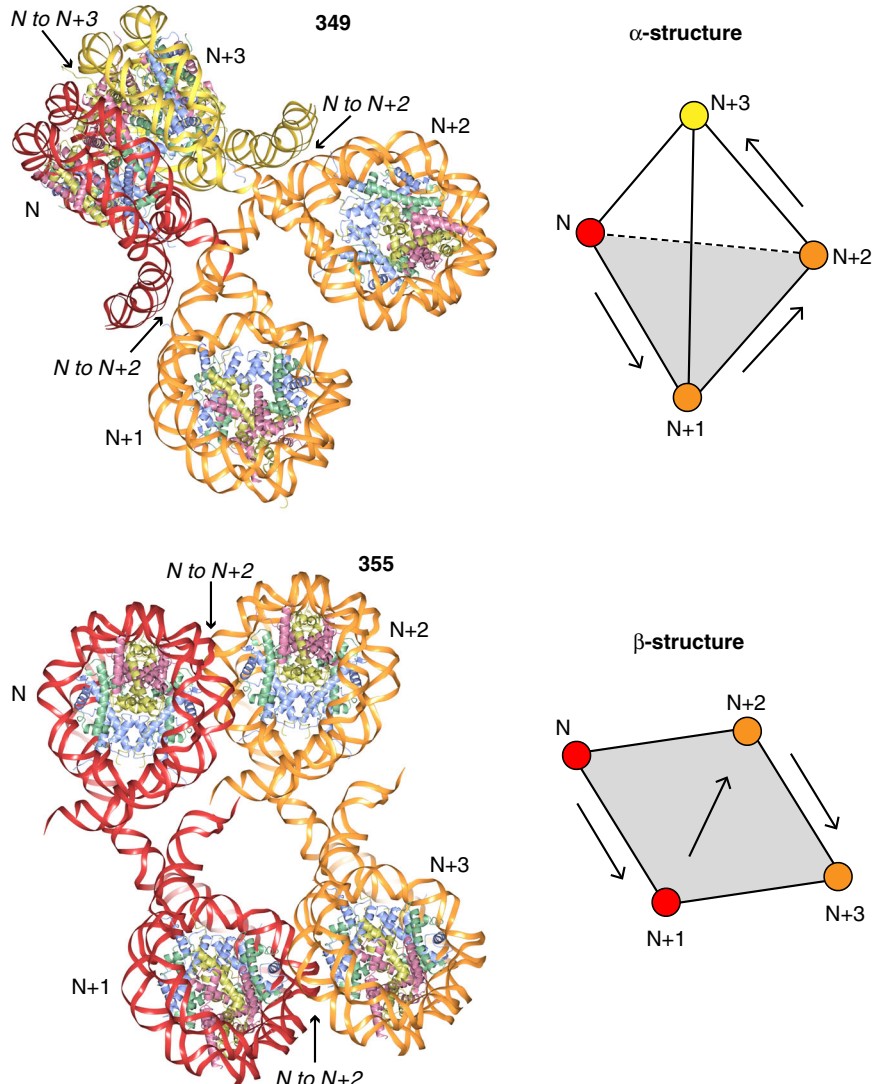

**Fig. 7 Agreement between nucleosome fibre structures and nucleosomal proximity mapping studies of chromatin.** Four-nucleosome sections of the 349 and 355 fibres are shown on the left, with the double helix coloured by dinucleosome repeat (as shown in Fig. 2). Nucleosome-nucleosome proximity relationships (primarily N to N + 2, secondarily N to N + 3) found to be enriched in chromatin mapping studies[19,24,29] are indicated. On the right, for comparison, is a schematic representation of the folding motifs found in yeast chromatin[24].

stretched out (and untwisted) the 2-start 30 nm fibre (272 Å height, 272 Å width and 287 Å length over 12 nucleosomes) so as to largely unfold it. In summary, the structures and approach presented here may help to provide a more unified understanding of local chromatin organization and dynamics as well as offer a means for further atomic-level characterization of nucleosome compaction and architectural factor activities.

## Methods

**Production of DNA fragments**. The non-palindromic 349 and 355 DNA fragments consist respectively of 345 bp and 351 bp duplexes with 4-nucleotide overhangs at each 3′ terminus (Fig. 1). Three tandem repeats of the 349 and 355 DNA fragments were cloned into the pUC19 vector with KpnI sites flanking each repeat (GenScript, Piscataway, NJ, USA). Subsequent to bacterial overexpression and plasmid preparation, the DNA fragments were cleaved from the vector by digestion with KpnI. This was followed by FPLC purification using a Resource Q column (GE Healthcare, Chicago, IL, USA) to remove vector. The sequences of both DNA strands are as follows:

349-Strand 1
CGCTGGAAAAAAAAAAACGCATCCCGGTGCCGAGGCCGCTCAATTGGT
CGTAGACAGCTCTAGCACCGCTTAAACGCACGTACGCGCTGTCTACCGC
GTTTTAACCGCCACTAGAAGCGCTTACTAGTCTCCAGGCACGTGTGAGA
CCGGCACATGAAAAAAAAAAGCATGCTCGAGTATGAAAAAAAAAAACGC

349-Strand 2
CGCTGGTTTTTTTTTTCATGTGCCGGTCTCACACGTGCCTGGAGACTA
GTAAGCGCTTCTAGTGGCGGTTAAAACGCGGTAGACAGCGCGTACGTG
CGTTTAAGCGGTGCTAGAGCTGTCTACGACCAATTGAGCGGCCTCGGCA
CCGGGATGCGTTTTTTTTTTCATACTCGAGCATGCTTTTTTTTTTCATGTG
CCGGTCTCACACGTGCCTGGAGACTAGTAAGCGCTTCTAGTGGCGGTTA
AAACGCGGTAGACAGCGCGTACGTGCGTTTAAGCGGTGCTAGAGCTGT
CTACGACCAATTGAGCGGCCTCGGCACCGGGATGCGTTTTTTTTTTCCA
GCGGTAC

355-Strand 1
CGCTGACGAAAAAAAAAAACGCATCCCGGTGCCGAGGCCGCTCAATTG
GTCGTAGACAGCTCTAGCACCGCTTAAACGCACGTACGCGCTGTCTACC
GCGTTTTAACCGCCACTAGAAGCGCTTACTAGTCTCCAGGCACGTGTGA
GACCGGCACATGAAAAAAAAAATGCATGCTCGAGTATGAAAAAAAAAA
ATCGCATCCCGGTGCCGAGGCCGCTCAATTGGTCGTAGACAGCTCTAGC
ACCGCTTAAACGCACGTACGCGCTGTCTACCGCGTTTTAACCGCCACTA
GAAGCGCTTACTAGTCTCCAGGCACGTGTGAGACCGGCACATGAAAAA
AAAAACGTCAGCGGTAC

355-Strand 2
CGCTGACGTTTTTTTTTTCATGTGCCGGTCTCACACGTGCCTGGAGA
CTAGTAAGCGCTTCTAGTGGCGGTTAAAACGCGGTAGACAGCGCGTACG

TGCGTTTAAGCGGTGCTAGAGCTGTCTACGACCAATTGAGCGGCCTCGG
CACCGGGATGCGATTTTTTTTTTTCATACTCGAGCATGCATTTTTTTTTT
CATGTGCCGGTCTCACACGTGCCTGGAGACTAGTAAGCGCTTCTAGTG
GCGGTTAAAACGCGGTAGACAGCGCGTACGTGCGTTTAAGCGGTGCTA
GAGCTGTCTACGACCAATTGAGCGGCCTCGGCACCGGGATGCGTTT
TTTTTTTCGTCAGCGGTAC

**Linker histone production.** *Homo sapiens* H1.0 was cloned into the pET-15b vector with an N-terminal hexa-histidine tag (EZBiolab Inc., Carmel, IN, USA). Linker histone was expressed in BL21(DE3) or BL21(DE3)pLysS *Escherichia coli* cells cultured in 2xTY medium at 37 °C. Subsequent to the cell density reaching an $OD_{600}$ of 0.6, protein expression was induced by adding IPTG (isopropyl β-D-1-thiogalactopyranoside) to a final concentration of 0.5 mM and allowed to proceed for 3–4 h at 37 °C or overnight at 18 °C. Cells were harvested by centrifugation, and the pellets resuspended in lysis buffer (50 mM Tris-HCl [pH 8.0], 2% Triton X-100, 0.05% [v/v] Calbiochem protease inhibitor cocktail set III—EDTA-free [EMD Millipore, Billerica, MA, USA]) with 4 mg/ml lysozyme, followed by stirring at 4 °C for 1 h. Subsequent to centrifugation at 4 °C, the pellet was resuspended in lysis buffer containing 1 M NaCl. After sonication on ice, the solution was allowed to incubate with stirring at 4 °C overnight. Subsequent to centrifugation at 4 °C, the supernatant, which contains linker histone, was retained.

Linker histone was purified starting with FPLC affinity-tag purification using a HisTrap HP column (GE Healthcare, Chicago, IL, USA). Heavy DNA contamination was removed by washing the protein-loaded HisTrap column with 2 M NaCl prior to elution. The fractions containing linker histone were pooled and further purified with a HiTrap Heparin HP column (GE Healthcare, Chicago, IL, USA). Linker histone-rich fractions were subsequently pooled and the N-terminal His-tag was removed by the addition of human rhinovirus 3C protease (Thermo Fisher Scientific, Waltham, MA, USA). The His-tag-cleaved linker histone sample was subjected to another round of HisTrap purification, followed by Heparin purification, to remove the uncleaved His-tagged linker histone and protease, as well as any degraded protein. Linker histone was concentrated and stored in 20 mM K-cacodylate [pH 6.0] at −80 °C. The molecular weight of the purified linker histone was determined by mass spectrometry analysis, which confirmed the full length nature of H1.0.

**Dinucleosome and linker histone complex assembly.** Nucleosomal constructs and linker histone complexes were assembled with recombinant *H. sapiens* core histones[44,45] and either the 349 or 355 DNA fragment. The core histones with N-terminal hexa-histidine tags were expressed in BL21(DE3) (H2A, H2B and H3) or JM109(DE3) (H4) *E. coli* cells and affinity purified with a HiTrap IMAC FF column (GE Healthcare, Chicago, IL, USA). The N-terminal His-tag was removed by thrombin digestion, and histones were purified further by FPLC using Resource S (H2A, H2B and H3) or Mono S (H4) columns (GE Healthcare, Chicago, IL, USA). Dinucleosomes were reconstituted based on established protocols[46] and shown to be capable of assembling into long fibres in solution (Supplementary Fig. 5). To generate linker histone assemblies, linker histone was mixed with nucleosome in molar stoichiometry ensuring saturation (Supplementary Fig. 6).

**Dinucleosome ligation assay.** In total, 349 and 355 dinucleosome (2 μM) samples were ligated using 40 Units of T4 ligase (NEB, Ipswich, Massachusetts, USA) in ligation buffer containing 50 mM Tris (pH 7.5), 10 mM $MgCl_2$, 1 mM adenosine-5′-triphosphate and 10 mM dithiothreitol in a 50 μl reaction volume. The samples were incubated at room temperature, and for each time point (0.5, 1, 2 or 3 h) 10 μl sample were withdrawn. The reaction was stopped with 25 mM EDTA followed by subjection to 0.8% agarose gel electrophoresis (using a running buffer [0.25X TBE] of 22.3 mM Tris, 22.5 mM boric acid and 0.5 mM EDTA). Gels were stained with ethidium bromide for DNA visualization. The procedure described above was repeated with four independent experiments; a representative result in shown in Supplementary Fig. 5.

**Crystallization and data collection.** Assemblies were prepared for crystallization by mixing linker histone and dinucleosome with a slight excess of linker histone (1.2–1.4 linker histone:nucleosome molar stoichiometry) in a buffer consisting of 80–100 mM $CaCl_2$, 50 mM KCl and 20 mM Na-acetate [pH 4.5] for 349 and 40–80 mM $CaCl_2$, 50 mM KCl and 20 mM Na-acetate [pH 4.5] for 355 to give a final total concentration of 4 mg ml⁻¹. Dinucleosome was prepared for crystallization in the same fashion by omitting linker histone. Crystals of linker histone assemblies and dinucleosome were grown by the hanging-droplet vapour diffusion method through salting-in via equilibration against a reservoir solution containing 40–50 mM $CaCl_2$, 25 mM KCl and 10 mM Na-acetate [pH 4.5] for 349 and 20–40 mM $CaCl_2$, 25 mM KCl and 10 mM Na-acetate [pH 4.5] for 355, with incubation at 18 °C.

Linker histone assemblies and dinucleosome crystals were harvested and stabilized in a buffer consisting of 10–20 mM $CaCl_2$, 12.5 mM KCl, 10 mM Na-acetate [pH 4.5], 25% 2-methyl-2,4-pentanediol (MPD) and 2% trehalose. For testing X-ray diffraction quality, the MPD concentration of the stabilization buffer was increased gradually up to 65% prior to data set collection, which for some of the crystal systems yielded a pronounced gain in diffraction quality. Structures are reported from crystals stabilized in both 25 and 65% MPD (i.e., at low and high cryoprotectant concentration). Single crystal X-ray diffraction data were recorded, subsequent to mounting stabilized crystals directly into the cryocooling $N_2$ gas stream set at −175 °C (ref. [47]), at beam line X06DA of the Swiss Light Source (Paul Scherrer Institute, Villigen, Switzerland) using a Pilatus 2M-F detector and an X-ray wavelength of 1.0 Å. For 349 crystals grown in the absence of linker histone, data was collected using a Rigaku FR-X ultra high-intensity microfocus rotating anode X-ray generator with a Pilatus 300 K detector and an X-ray wavelength of 1.54 Å. Data collection statistics are given in Tables 1–3.

**Structure solution, refinement and analysis.** Diffraction data were indexed, integrated, merged, scaled and evaluated with a combination of iMosflm[48,49], XDS[50], autoPROC[51], SCALA[52] and AIMLESS[53] from the CCP4 package[54,55] and in-house data processing pipelines, *go.com* and *go.py*, developed by the Swiss Light Source macromolecular crystallography beamlines (Paul Scherrer Institute, Villigen, Switzerland).

Initial phases for solving structures were obtained by molecular replacement using the programme PHASER[56] from the CCP4 package[54,55], with the 2.2 Å resolution crystal structure of NCP composed of the 601L DNA fragment[38] (*pdb* code 3UT9) serving as the search model. The avian H5 (a.k.a. avian H1.0) globular domain from the assembly with a 167 bp nucleosome crystal structure[31] (*pdb* code 4QLC) was used as a linker histone search model in molecular replacement. For 349 dinucleosome assembled with linker histone, the two data sets collected at 25 and 65% MPD both show a common site of linker histone association in the non-dyad mode. For the 65% MPD data, an additional linker histone binding site, corresponding to the on-dyad mode, is evident.

Atomic refinement and model building were carried out with REFMAC[57] and COOT[58], respectively, from the CCP4 suite[54,55]. Structure refinement statistics are given in Tables 1–3. Double helix conformational parameters were obtained with 3DNA[59,60]. Graphic figures were prepared with PyMOL (DeLano Scientific LLC, San Carlos, CA, USA) and CCPmg[61], and movies were made using PyMOL.

**Reporting summary.** Further information on research design is available in the Nature Research Reporting Summary linked to this article.

## Data availability
Atomic coordinates and structure factors for the four highest resolution 349 and 355 models have been deposited in the Protein Data Bank under accession codes 6LA8, 6LA9, 6M44 and 6M3V respectively. Data supporting the findings of this work are available within the article and Supplementary Information files. The 349 and 355 DNA constructs unique to this study are available through Addgene (www.addgene.org; ID-159643 and ID-159644). All other data are available from the corresponding author upon request.

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

## Acknowledgements

We thank V. Olieric, M. Wang, and staff at the Swiss Light Source (Paul Scherrer Institute, Villigen, Switzerland). We are grateful to W.K. Shum, S. Ravi, Z. Ma, S. Padavattan and G.E. Davey for their assistance on the project. We thank Q. Bao for helping to lay the foundation for this work and L. Nordenskiöld for his support. This project was funded by the Singapore Ministry of Education Academic Research Fund Tier 1 (grants 2014-T1-001-049 and 2017-T1-002-020), Tier 2 (grant MOE2015-T2-2-089) and Tier 3 (grant MOE2012-T3-1-001) Programmes. The research leading to these results has received funding from the European Union's Horizon 2020 research and innovation programme under grant agreement #730872, project CALIPSOplus. C.A.D. dedicates this work to the memory of Agnieszka Marie Mordas Martin.

## Author contributions

Investigation and methodology, Z.A., D.S and P.L.L.; Data curation, Z.A., D.S., and P.L.L.; Analysis, validation and visualization, Z.A., D.S., P.L.L. and C.A.D.; Conceptualization, funding acquisition, supervision and writing−original draft, C.A.D.; Writing−review and editing, Z.A., D.S. and C.A.D.

## Competing interests

The authors declare no competing interests.
