## [Peer Review File · Nature Communications]

REVIEWER COMMENTS

Reviewer #1 (Remarks to the Author):

How chromatin condenses to form higher-order structures is an important but unsolved problem. In this manuscript, the authors report the crystal structures of dinucleosomes with cohesive-ends that were used to connect the dinucleosomes to mimic the continuous double helix DNA. The packing of the di-nucleosomes in the crystal suggests that chromatin might condense to form higher-order structures with open zigzagged conformation of nucleosomes that interdigitate between neighboring chromatin chains (or distant nucleosomes). This type of nucleosome packing is very different from the conventional view that chromatin condenses to high-order structures through formation of the so-called 30-nm fibers.

Although it has been proposed that chromatin or nucleosome arrays may form condensed structures through interdigitation of nucleosomes, models with structural details are lacking. The crystal structures presented in this manuscript provided physically more realistic models for inter-chromatin chain interactions, although whether they represent the structures of the condensed chromatin in vivo remains to be confirmed. The work represents a significant advance in structural studies of chromatin in its condensed form.

Following issues need to be clarified before it is accepted for publication:

- (1) How are 349 and 355 bp DNA for the dinucleosome chosen? In Figure 1, are the two structures models or real structures?
- (2) The paired DNA region in the crystal structure needs to be shown in an enlarged image and compared with normal B-DNA conformation to show how well it mimics the B-DNA.
- (3) Interactions of nucleosomes need to be shown more clearly. The residues involved in the interactions should be labeled.
- (4) Are the tails of histone tails observed in the crystal structures? If so, what do they interact with? Are they involved in nucleosome packing?
- (5) The on-dyad binding of the chromatosome should be compared with the known structure of the mono-chromatosome containing the globular domain of H5 (reference 31).
- (6) The observation of nucleosome fiber density increases with dehydration (Figure S1) is interesting. It suggests that the nucleosome arrays exist as a dynamic structure in solution. The authors should highlight the changes of the conformations due to dehydration.
- (7) Is there any space in the crystals that would allow HP1 binding to the nucleosome? The cryo-EM structure of HP1 bound to the nucleosome is available known (Machida et al. Mol Cell, 2018).

Reviewer #2 (Remarks to the Author):

This manuscript uses x-ray crystallography to determine the structure of the fibrous assemblies formed by self-association of defined dinucleosomes in high concentrations of divalent cations. The assemblies have an open zig-zag structure that permits interdigitation of one assembly with another. Linker DNA length is shown to produce specific differences in the structure of the assemblies. Both nucleosomal and no-nucleosomal modes of interaction histone H1 are observed. From a crystallography perspective the work represents a technical tour de force. Overall, the results reported here have the potential to make an important contribution to understanding how chromatin is packaged within chromosomes

Comments

1. The importance of this work rests with its physiological relevance. Are these open zig-zag interdigitated fibers at all related to chromatin structures that form in the cell? I believe they are, but this has not been properly conveyed in the manuscript and as such the paper lacks important context

in its present form. On page 3, paragraph 2 the authors introduce the subject of chromatin dynamics. Reference is made to the folded 30-nm structures seen in vitro in low salt and to the interdigitated packaging seen by some in chromosomes. However, no mention is made of what happens to chromatin in vitro under the high divalent cation conditions (40-100 mM Ca²⁺) used in the present studies. It has long been known that endogenous chromatin fragments and reconstituted nucleosomal arrays self-associate under these ionic conditions into large structures that have been called precipitants, oligomers, or condensates. Importantly, the chromatin within the condensates is extended rather than folded into 30-nm fibers and is thought to be packaged as interdigitated zig-zags (ref. 10). In other words, contiguous arrays of nucleosomes self-assemble under high divalent cation conditions into biologically relevant structures that have been proposed to be remarkably similar to the self-assembled structures formed by dinucleosomes reported here by Adhireskan et al. Moreover, the total divalent cation concentration (Ca²⁺/Mg²⁺) at the surface of mitotic chromosomes is 30-50 mM (Strick R et al., J. Cell Biol, 2001). Altogether, that makes the present results extremely important and biologically pertinent. Consequently, I recommend that the in vitro chromatin self-association information be integrated into the paper to justify the physiological relevance of the dinucleosome assemblies, both in the Introduction as well as the Discussion.

2. As the authors themselves point out in the Discussion, chromatin in vivo does not have perfectly regular linker DNA lengths. In this context, what are the ramifications of the findings that the detailed structure of the 349 and 355 constructs and their modes of interdigitation are different? Specifically, how do the authors propose that mixed linker chromatin might interdigitate in light of the model study results? Would mixed linker chromatin even be able to self-assemble into interdigitated structures? Again, this bears on the in vivo significance of the dinucleosome assemblies.

3. In several places the authors refer to the histone-histone and histone-DNA contacts within the assemblies and cite Fig. 2B. If possible, it would be nice to add some details of these contacts. It is not clear to me where and what these contacts are by looking at Fig. 2B and reading the accompanying legend.

4. I find the non-nucleosomal binding mode of H1 to be novel and potentially quite interesting. Could this finding perhaps be discussed more completely?

5. The text/numbers in all the figures are too small, especially if the figures get reduced for publication.

Reviewer #3 (Remarks to the Author):

The authors of this manuscript describe the crystal structures of two dinucleosomes with different DNA lengths (349 and 355 bp dinucleosomes) in the presence (and absence) of linker histone H1.0 at near atomic resolution. These dinucleosomes have been designed to have cohesive DNA ends in such a way that they form continuous fibers that could be observed in the crystals. This allows the study of the compaction of these fibers and helps to improve the resolution of the crystals. They have also improved the resolution of their crystals by increasing the amount of cryoprotectant used prior data collection, creating a dehydration effect.

In both 349 and 355 fibers structure, zigzag conformations of the nucleosomes have been observed with strong interdigitation between the nucleosomes of neighboring symmetric fibers. However, the slight difference in DNA linker lengths between the 349 and 355 fibers (2 additional bp in the 'shared' DNA arm that links the nucleosomes of the same dinucleosome and 4 bp in the 'paired' linker DNA that connects dinucleosomes in the fiber), which in 355 coincides with double helix twist extensions relative to the 349 fiber in the shared and paired linker DNA, induces a rotational effect and bending of linker DNA which strongly affects nucleosome packing. From these observations the authors

conclude that linker DNA lengths (which introduce twist differences) have a dominant effect in fiber configuration. While the 355 fiber shows a more extended fiber, dinucleosome repeats perpendicular to the fiber axis, nucleosomes stacked edgewise creating a double-sided staircase, shared linker DNA almost orthogonal to the fiber axis, and paired linker DNA between dinucleosomes almost coincident with the fiber axis, the 349 fiber shows a very different and less extended zigzag configuration. The 349 fiber axis runs between the dinucleosome repeats which stack in an off-set fashion, showing near planar nucleosomes angled along the fiber axis and each dinucleosome paired with two dinucleosomes of the opposing column. The interaction between nucleosomes in the 355 fiber models are N+2 (interactions between nucleosomes N and N+2) while in the 349 fiber model are N+2 as well as N+3 (interactions between nucleosomes N and N+3).

With respect to the fibers assembled with the linker histone H1.0, the authors did not observe differences between the fiber conformations in presence or absence of the linker histone. Only the globular domain of H1.0 could be observed in the 349 fibers (not visible in 355-H1.0 fibers due to disorder induced by mixed binding modes). Two different crystal structures were obtained from the 349-fiber assembled using low (L) and high stoichiometric (H) ratios of H1.0 (structures obtained at 3.4Å and 3.7Å, respectively). In the structure of the 349-H1.0-L fiber they observed the globular domain of H1.0 located only in an unusual 'non-dyad' position (interacting with the DNA of three neighbouring nucleosomes cores of the interdigitated fibers) thus mediating inter-fiber contacts which would help consolidate the compacted structure. The structure of 349-H1.0-H shows the globular domain of H1.0 in the 'non-dyad' localization but also in the already described 'on-dyad' position, binding the dyad of one of the nucleosomes of the dinucleosome (intra-nucleosomal contacts) as well as the 'shared' linker DNA of a neighbouring dinucleosome, in close proximity due to the dense packing (inter-fiber contacts).

Finally, the authors conclude that nucleosomes have a strong tendency to interdigitate but differences on DNA linker length will determine specific condensing features, many fiber configurations and variations in linker histone binding. These different fiber arrangements observed in their structures are in agreement with the observations of local chromosome architecture and the heterogeneities of the chromatin structure observed recently with other techniques.

This work which is focused on the structural analysis of fibers observed by X-ray crystallography reinforces the idea of chromatin versatility (particularly on the influence of nucleosome linker lengths) and aims to explain at near atomic level the observations of chromatin heterogeneities in vivo and that do not conform with the classic 30nm fiber models.

The structure of different arrays of nucleosomes published in recent years also point to this variability of chromatin structure under different conditions (DNA linker length, ion strength, chromatin binding factors, histone variants, etc.). The observation of long 'fibers' in the crystals presented in this work adds to this knowledge. However, their presence should also be proven in solution (e.g.: images of cryo-EM microscopy, analytical centrifugation (variation on sedimentation velocities during the formation of the fibers), SAXS, etc.). The use of some of these techniques (or other found appropriate by the authors) will convince and reinforce the existence of such fibers and discard the strong effects of crystal packing.

A novel claim in this paper is the 'non-dyad' binding of linker histone H1.0 observed in the 349-H1.0-L and 349-H1.0-H fibers. While the role of linker histones in packing/compaction of DNA seems well established, their binding modes to the nucleosomes might be related not only to linker histone variants but also to the compaction state of the nucleosomal arrays (an extended/intermediate chromatin configuration could favor an 'on-dyad' binding mode while a more compacted fiber (30nm fiber) might have an 'off-dyad' binding mode), however the 'non-dyad' binding mode, far away from the dyad of the nucleosome, has not been described. According to the authors, this type of binding mode observed in the globular domain of histone H1.0 would help in stabilizing inter-fiber connectivity. However, they should discard that this binding mode is not the result of an artifact during the reconstitution of the dinucleosome-linker histone H1.0 complex. H1.0 linker histone might 'prefer' snuggling between 3 external nucleosomes (non-specific binding), instead that binding to the nucleosome dyad which is less accessible in the fiber. This suspicion is reinforced by the observation

that only when there is an excess of H1.0 on the 349-H1.0-H fiber, the 'on-dyad' binding mode is observed. To address this question this binding mode should be shown biochemically (for e.g. DNA footprinting experiments, mutations of the residues of H1.0 involved in the 'non-dyad' binding mode (or other)).

Summarizing: the presence of the fibers should be proven in solution and the 'non-dyad' binding mode of H1.0 in the 349 fibers should be addressed to convince the reader of this unusual binding, discarding any possible non-specific binding or artifacts introduced by the fiber configuration during the reconstitution of the complex.

This manuscript is not acceptable in its present form, but authors are strongly encouraged to consider a resubmission in the future when the main points that will support their crystal structures will be addressed.

Other remarks to the manuscript:

Manuscript in general is concise and clearly written.

- Concerning the description of the fibers in the text (pages 5-6), a figure specifying the fibers' dimensions would be appreciated.

- In Figure 2B, could the Ca²⁺ ions be labeled?. It is difficult to distinguish the dash black and purple lines (maybe due to the size of the picture in the manuscript).

In general, a video showing the 349 and 355 fibers, as described in figure 2 and 3, will be most explanatory for readers.

- A more detailed description of the interdigitation interfaces between nucleosomes in the 349 and 355 fibers would be of interest: which histones are involved in these histone-histone and histone-DNA interactions?. This could lead to a discussion if PTMs or histone variants could have an effect in the formation of these type of fibers and open further investigations.

- Page 11: The authors mention in the manuscript 'Importantly, the fiber models provide a means to fit/interpret a proximity map of human cellular chromatin²⁹ without the need for invoking folded helical 30 nm structures.' How these two observed fibers 349 and 355 fit one of these published results?. Could the authors be more specific and show the comparison?.

- Page 12: An extra figure (in supplementary materials?) showing the comparison between the 355 fiber structure and the 2-start zigzag form of 30nm structure would be of interest to support the similarities described in the text.

- Figure 1: The description of the cohesive DNA ends in the text and in the figure are confusing.

- Figure 5: An omit map of the globular domain of H1.0 in the 349-H1.0-L and -H structures will be needed.

- Page 21 (Methods): With respect to the overnight incubation of the lysate during the purification of H1.0, why is this incubation required?. Is there not a risk of H1.0 degradation?.

- Could the speed of centrifugations described in the Methods be specified?.

- Page 22: An SDS-page showing the reconstitution and level of occupancy of H1 in the dinucleosomes complexes would be of interest.

Structure refinements:

- Could the authors comment on the use of Rfree test size of 2% during their structure refinements? (which usually are between 5% or 10%).

RESPONSES TO REVIEWER COMMENTS

We thank the reviewers wholeheartedly for their valuable time and input on our manuscript. We have carefully considered all of the recommendations of the reviewers, whose insightful comments have been very helpful in compiling the revised version. Below we outline the revisions implemented for each of the points raised by the reviewers.

In response to comments by Reviewer #1:

How chromatin condenses to form higher-order structures is an important but unsolved problem. In this manuscript, the authors report the crystal structures of dinucleosomes with cohesive-ends that were used to connect the dinucleosomes to mimic the continuous double helix DNA. The packing of the di-nucleosomes in the crystal suggests that chromatin might condense to form higher-order structures with open zigzagged conformation of nucleosomes that interdigitate between neighboring chromatin chains (or distant nucleosomes). This type of nucleosome packing is very different from the conventional view that chromatin condenses to high-order structures through formation of the so-called 30-nm fibers.

Although it has been proposed that chromatin or nucleosome arrays may form condensed structures through interdigitation of nucleosomes, models with structural details are lacking. The crystal structures presented in this manuscript provided physically more realistic models for inter-chromatin chain interactions, although whether they represent the structures of the condensed chromatin in vivo remains to be confirmed. The work represents a significant advance in structural studies of chromatin in its condensed form.

Following issues need to be clarified before it is accepted for publication:

1. *How are 349 and 355 bp DNA for the dinucleosome chosen? In Figure 1, are the two structures models or real structures?*

The 349 and 355 constructs were designed with the intent of acquiring a system that can self-assemble in an open-ended, ‘fibrous’ fashion, as opposed to forming small, closed, circular structures. The 349 and 355 structures shown in Figure 1 are from the highest resolution models in their class. These are the asymmetric units, and this has been clarified in the figure legend.

2. *The paired DNA region in the crystal structure needs to be shown in an enlarged image and compared with normal B-DNA conformation to show how well it mimics the B-DNA.*

This has been incorporated as Supplementary Figure 2.

3. *Interactions of nucleosomes need to be shown more clearly. The residues involved in the interactions should be labeled.*

These residues are now labelled in the relevant figures.

4. *Are the tails of histone tails observed in the crystal structures? If so, what do they interact with? Are they involved in nucleosome packing?*

We have included a discussion of this in paragraph 3 of the Discussion section.

5. *The on-dyad binding of the chromatosome should be compared with the known structure of the mono-chromatosome containing the globular domain of H5 (reference 31).*

Panel c in Figure 6 has been added, which shows this comparison.

6. *The observation of nucleosome fiber density increases with dehydration (Figure S1) is interesting. It suggests that the nucleosome arrays exist as a dynamic structure in solution. The authors should highlight the changes of the conformations due to dehydration.*

An additional panel (a) has been added to Supplementary Figure 1 to highlight the conformational differences between hydration levels.

7. *Is there any space in the crystals that would allow HP1 binding to the nucleosome? The cryo-EM structure of HP1 bound to the nucleosome is available known (Machida et al. Mol Cell, 2018).*

Given the dense packing of nucleosomes in both the 349 and 355 fiber structures, it is not obvious how HP1 would be accommodated with exactly the same presiding configurations in each case. However, as we noted in the discussion, variations in histone composition and the presence of architectural factors, in addition to the activity of other cationic species, would be expected to have a significant impact on fiber structure or dynamics in a localized fashion. This would help account for the observed heterogeneity in compact chromatin structures.

In response to comments by Reviewer #2:

This manuscript uses x-ray crystallography to determine the structure of the fibrous assemblies formed by self-association of defined dinucleosomes in high concentrations of divalent cations. The assemblies have an open zig-zag structure that permits interdigitation of one assembly with another. Linker DNA length is shown to produce specific differences in the structure of the assemblies. Both nucleosomal and no-nucleosomal modes of interaction histone H1 are observed. From a crystallography perspective the work represents a technical tour de force. Overall, the results reported here have the potential to make an important contribution to understanding how chromatin is packaged within chromosomes.

Comments:

8. *The importance of this work rests with its physiological relevance. Are these open zig-zag interdigitated fibers at all related to chromatin structures that form in the cell? I believe they are, but this has not been properly conveyed in the manuscript and as such the paper lacks important context in its present form. On page 3, paragraph 2 the authors introduce the subject of chromatin dynamics. Reference is made to the folded 30-nm structures seen in vitro in low salt and to the interdigitated packaging seen by some in chromosomes. However, no mention is made of what happens to chromatin in vitro under the high divalent cation conditions (40-100 mM Ca²⁺) used in the present studies. It has long been known that endogenous chromatin fragments and reconstituted nucleosomal arrays self-associate under these ionic conditions into large structures that have been called precipitants, oligomers, or condensates. Importantly, the chromatin within the condensates is extended rather than folded into 30-nm fibers and is thought to be packaged as interdigitated zig-zags (ref. 10). In other words, contiguous arrays of nucleosomes self-assemble under high divalent cation conditions into biologically relevant structures that have been proposed to be remarkably similar to the self-assembled structures formed by dinucleosomes reported here by Adhireskan et al. Moreover, the total divalent cation concentration (Ca²⁺/Mg²⁺) at the surface of mitotic chromosomes is 30-50 mM (Strick R et al., J. Cell Biol, 2001). Altogether, that makes the present results extremely important and biologically pertinent. Consequently, I recommend that the in vitro chromatin self-association information be integrated into the paper to justify the physiological relevance of the dinucleosome assemblies, both in the Introduction as well as the Discussion.*

We have included a discussion of the *in vitro* nucleosome array condensate characterization in the Introduction (paragraph 3) and the Discussion (paragraph 1) sections.

9. *As the authors themselves point out in the Discussion, chromatin in vivo does not have perfectly regular linker DNA lengths. In this context, what are the ramifications of the findings that the detailed structure of the 349 and 355 constructs and their modes of interdigitation are different? Specifically, how do the authors propose that mixed linker chromatin might interdigitate in light of the model study results? Would mixed linker chromatin even be able to self-assemble into interdigitated structures? Again, this bears on the in vivo significance of the dinucleosome assemblies.*

We have included a discussion of this issue in paragraph 3 of the Discussion section.

10. *In several places the authors refer to the histone-histone and histone-DNA contacts within the assemblies and cite Fig. 2B. If possible, it would be nice to add some details of these contacts. It is not clear to me where and what these contacts are by looking at Fig. 2B and reading the accompanying legend.*

These residues are now labelled in the relevant figures.

11. *I find the non-nucleosomal binding mode of H1 to be novel and potentially quite interesting. Could this finding perhaps be discussed more completely?*

We have expanded the discussion of the linker histone binding modes in the Results section (last paragraph) and in the second to last paragraph of the Discussion section. We also introduced panels b and c of Figure 6 in support of this.

12. *The text/numbers in all the figures are too small, especially if the figures get reduced for publication.*

The font size has been increased for all of the figures.

In response to comments by Reviewer #3:

The authors of this manuscript describe the crystal structures of two dinucleosomes with different DNA lengths (349 and 355 bp dinucleosomes) in the presence (and absence) of linker histone H1.0 at near atomic resolution. These dinucleosomes have been designed to have cohesive DNA ends in such a way that they form continuous fibers that could be observed in the crystals. This allows the study of the compaction of these fibers and helps to improve the resolution of the crystals. They have also improved the resolution of their crystals by increasing the amount of cryoprotectant used prior data collection, creating a dehydration effect.

In both 349 and 355 fibers structure, zigzag conformations of the nucleosomes have been observed with strong interdigitation between the nucleosomes of neighboring symmetric fibers. However, the slight difference in DNA linker lengths between the 349 and 355 fibers (2 additional bp in the 'shared' DNA arm that links the nucleosomes of the same dinucleosome and 4 bp in the 'paired' linker DNA that connects dinucleosomes in the fiber), which in 355 coincides with double helix twist extensions relative to the 349 fiber in the shared and paired linker DNA, induces a rotational effect and bending of linker DNA which strongly affects nucleosome packing. From these observations the authors conclude that linker DNA lengths (which introduce twist differences) have a dominant effect in fiber configuration. While the 355 fiber shows a more extended fiber, dinucleosome repeats perpendicular to the fiber axis, nucleosomes stacked edgewise creating a double-sided staircase, shared linker DNA almost orthogonal to the fiber axis, and paired linker DNA between dinucleosomes almost coincident

with the fiber axis, the 349 fiber shows a very different and less extended zigzag configuration. The 349 fiber axis runs between the dinucleosome repeats which stack in an off-set fashion, showing near planar nucleosomes angled along the fiber axis and each dinucleosome paired with two dinucleosomes of the opposing column. The interaction between nucleosomes in the 355 fiber models are $N+2$ (interactions between nucleosomes N and $N+2$) while in the 349 fiber model are $N+2$ as well as $N+3$ (interactions between nucleosomes N and $N+3$).

With respect to the fibers assembled with the linker histone H1.0, the authors did not observe differences between the fiber conformations in presence or absence of the linker histone. Only the globular domain of H1.0 could be observed in the 349 fibers (not visible in 355-H1.0 fibers due to disorder induced by mixed binding modes). Two different crystal structures were obtained from the 349-fiber assembled using low (L) and high stoichiometric (H) ratios of H1.0 (structures obtained at 3.4\AA and 3.7\AA , respectively). In the structure of the 349-H1.0- L fiber they observed the globular domain of H1.0 located only in an unusual 'non-dyad' position (interacting with the DNA of three neighbouring nucleosomes cores of the interdigitated fibers) thus mediating inter-fiber contacts which would help consolidate the compacted structure. The structure of 349-H1.0- H shows the globular domain of H1.0 in the 'non-dyad' localization but also in the already described 'on-dyad' position, binding the dyad of one of the nucleosomes of the dinucleosome (intra-nucleosomal contacts) as well as the 'shared' linker DNA of a neighbouring dinucleosome, in close proximity due to the dense packing (inter-fiber contacts).

Finally, the authors conclude that nucleosomes have a strong tendency to interdigitate but differences on DNA linker length will determine specific condensing features, many fiber configurations and variations in linker histone binding. These different fiber arrangements observed in their structures are in agreement with the observations of local chromosome architecture and the heterogeneities of the chromatin structure observed recently with other techniques.

This work which is focused on the structural analysis of fibers observed by X-ray crystallography reinforces the idea of chromatin versatility (particularly on the influence of nucleosome linker lengths) and aims to explain at near atomic level the observations of chromatin heterogeneities *in vivo* and that do not conform with the classic 30nm fiber models.

13. The structure of different arrays of nucleosomes published in recent years also point to this variability of chromatin structure under different conditions (DNA linker length, ion strength, chromatin binding factors, histone variants, etc.). The observation of long 'fibers' in the crystals presented in this work adds to this knowledge. However, their presence should also be proven in solution (e.g.: images of cryo-EM microscopy, analytical centrifugation (variation on sedimentation velocities during the formation of the fibers), SAXS, etc.). The use of some of these techniques (or other found appropriate by the authors) will convince and reinforce the existence of such fibers and discard the strong effects of crystal packing.

We conducted a ligation analysis, which shows that both the 349 and 355 dinucleosomes can assemble into long fibres in solution. This is described in the Methods section and included as Supplementary Figure 5.

14. A novel claim in this paper is the 'non-dyad' binding of linker histone H1.0 observed in the 349-H1.0- L and 349-H1.0- H fibers. While the role of linker histones in packing/compaction of DNA seems well established, their binding modes to the nucleosomes might be related not only to linker histone variants but also to the compaction state of the nucleosomal arrays (an extended/intermediate chromatin configuration could favor an 'on-dyad' binding mode while a more compacted fiber (30nm fiber) might have an 'off-dyad' binding mode), however the 'non-dyad' binding mode, far away from the dyad of the nucleosome, has not been described.

According to the authors, this type of binding mode observed in the globular domain of histone H1.0 would help in stabilizing inter-fiber connectivity. However, they should discard that this binding mode is not the result of an artifact during the reconstitution of the dinucleosome-linker histone H1.0 complex. H1.0 linker histone might 'prefer' snuggling between 3 external nucleosomes (non-specific binding), instead that binding to the nucleosome dyad which is less accessible in the fiber. This suspicion is reinforced by the observation that only when there is an excess of H1.0 on the 349-H1.0-H fiber, the 'on-dyad' binding mode is observed. To address this question this binding mode should be shown biochemically (for e.g. DNA footprinting experiments, mutations of the residues of H1.0 involved in the 'non-dyad' binding mode (or other)).

Summarizing: the presence of the fibers should be proven in solution and the 'non-dyad' binding mode of H1.0 in the 349 fibers should be addressed to convince the reader of this unusual binding, discarding any possible non-specific binding or artifacts introduced by the fiber configuration during the reconstitution of the complex.

We note that the 349 and 355 fibre crystals were grown under just two distinct conditions—either in the complete absence of linker histone or with a saturating quantity of linker histone. This has been further clarified in the Results section. The non-dyad mode we observe at both 349 cryoprotectant concentrations, whereas the on-dyad mode is only observed with the high cryoprotectant concentration 349 data. We conducted an expanded analysis, which illustrates the similarity in DNA association between the on-dyad and non-dyad modes, rationalizing the occurrence of the latter (last paragraph of Results section and Figure 6b). We have also introduced further discussion of this issue in the second to last paragraph of the Discussion section, which includes a statement on in vivo relevance.

In any case, we note that an effective linker histone mapping study, which can address binding modes within interdigitated fibres, would require a special method for contending with aggregated/precipitated chromatin and have a level of sensitivity necessary to characterize potentially low or partial linker histone occupancies. This is beyond the scope of the present study. In this sense, it is also important to consider that most studies investigating linker histone binding mode have been carried out on mononucleosome substrates that would predispose uninnucleosomal association.

This manuscript is not acceptable in its present form, but authors are strongly encouraged to consider a resubmission in the future when the main points that will support their crystal structures will be addressed.

Other remarks to the manuscript:

15. *Manuscript in general is concise and clearly written.*

- Concerning the description of the fibers in the text (pages 5-6), a figure specifying the fibers' dimensions would be appreciated.

The fibre dimensions are now illustrated in Figure 2 and Figure 6.

16. *- In Figure 2B, could the Ca²⁺ ions be labeled?. It is difficult to distinguish the dash black and purple lines (maybe due to the size of the picture in the manuscript).*

The calcium ions and residues have been labelled, and the sizes of the figure panels have been increased.

17. *In general, a video showing the 349 and 355 fibers, as described in figure 2 and 3, will be most explanatory for readers.*

We made six movies to support all of the structural descriptions. These are included in the Supplementary Information.

18. - *A more detailed description of the interdigitation interfaces between nucleosomes in the 349 and 355 fibers would be of interest: which histones are involved in these histone-histone and histone-DNA interactions?. This could lead to a discussion if PTMs or histone variants could have an effect in the formation of these type of fibers and open further investigations.*

These residues are now labelled in the relevant figures. Potential effects from changes to these residues have been discussed in the Discussion section (paragraph 3).

19. - *Page 11: The authors mention in the manuscript 'Importantly, the fiber models provide a means to fit/interpret a proximity map of human cellular chromatin²⁹ without the need for invoking folded helical 30 nm structures.' How these two observed fibers 349 and 355 fit one of these published results?. Could the authors be more specific and show the comparison?.*

We have expanded the discussion of this in the last paragraph of the Discussion section. In addition, Figure 7 has been added in support of the analysis.

20. - *Page 12: An extra figure (in supplementary materials?) showing the comparison between the 355 fiber structure and the 2-start zigzag form of 30nm structure would be of interest to support the similarities described in the text.*

To our knowledge, a structure for the 2-start, zigzag 30 nm model was never released to the public (Song et al., 2014; Chen et al., 2014; Li & Zhu, 2015). We have therefore now added an elaboration in the Discussion (last paragraph) with some additional geometric comparisons between the 355 and the 2-start 30 nm structures.

21. - *Figure 1: The description of the cohesive DNA ends in the text and in the figure are confusing.*

We have clarified the description in the text and in Figure 1. A supplementary figure (2) has also been added, which additionally illustrates the cohesive end annealing.

22. - *Figure 5: An omit map of the globular domain of H1.0 in the 349-H1.0-L and -H structures will be needed.*

Omit maps supporting the on-dyad and non-dyad linker histone binding observed are included as Supplementary Figures 3 and 4.

23. - *Page 21 (Methods): With respect to the overnight incubation of the lysate during the purification of H1.0, why is this incubation required?. Is there not a risk of H1.0 degradation?. - Could the speed of centrifugations described in the Methods be specified?.*

The 4 deg. overnight incubation with stirring is to promote as much solubilisation of H1.0 as possible, to maximize yields. A protease inhibitor cocktail was included to suppress degradation. The centrifugation speeds have been added in the Methods.

24. - *Page 22: An SDS-page showing the reconstitution and level of occupancy of H1 in the dinucleosomes complexes would be of interest.*

Accurate concentrations for linker histone were confirmed by titrations against mononucleosome. To ensure saturation, slight molar excesses of H1 were used for the crystallization reconstitutions. While SDS-PAGE analysis would just reflect the input H1 concentration, we have instead included a band-shift gel representing an H1.0-nucleosome titration (Supplementary Figure 6).

25. *Structure refinements:*

- *Could the authors comment on the use of R_{free} test size of 2% during their structure refinements? (which usually are between 5% or 10%).*

We attempt to utilize as many working reflections as possible without compromising the statistical accuracy of the free R value (A.T. Brünger, 1997, *Methods Enzymol.*, **277**: 366-396). For the deposited fibre models, even for the lowest resolution structure, this corresponds to a minimum of >500 reflections in the free R set.

REVIEWERS' COMMENTS:

Reviewer #1 (Remarks to the Author):

I have read the response of the authors to the earlier comments and the revised manuscript. The revised manuscript is improved substantially. I recommend its publication in Nature Communications without the need of further revision.

Reviewer #2 (Remarks to the Author):

None

Reviewer #3 (Remarks to the Author):

The revised version of the manuscript includes the suggestions and addresses the issues that were raised. This version is now more comprehensive, explanatory and clear and it is considered acceptable for publication.

Minor remarks:

Comments to Point 20: The mention to a 2-start zigzag 30nm 'model' referred to the 2-start zig-zag 30 'form' that is mentioned in the manuscript and which points to reference: Song et al. (2014) Science, 344, 376-380. The new elaboration in the Discussion is precisely what was asked.

Figure 3: Concerning the intrafibre nucleosome-nucleosome interactions within the 349 fibre, the side chain of H3 Q76 of NUC 5 is not visible (is only CA shown?).

Figure 6 c: The color of the H5-NUC structure should be different than cyan for clarity since according to the authors cyan is the color used for the non-dyad linker histone (LH2) across the figures.

RESPONSES TO REVIEWER COMMENTS

We would like to thank the reviewers again for their valuable time and input on our manuscript.

Comments by Reviewer #1:

I have read the response of the authors to the earlier comments and the revised manuscript.

The revised manuscript is improved substantially. I recommend its publication in Nature Communications without the need of further revision.

Comments by Reviewer #2:

None

In response to comments by Reviewer #3:

The revised version of the manuscript includes the suggestions and addresses the issues that were raised. This version is now more comprehensive, explanatory and clear and it is considered acceptable for publication.

Minor remarks:

Comments to Point 20: The mention to a 2-start zigzag 30nm 'model' referred to the 2-start zig-zag 30 'form' that is mentioned in the manuscript and which points to reference: Song et al. (2014) Science, 344, 376-380. The new elaboration in the Discussion is precisely what was asked.

1. *Figure 3: Concerning the intrafibre nucleosome-nucleosome interactions within the 349 fibre, the side chain of H3 Q76 of NUC 5 is not visible (is only CA shown?).*

The view has been shifted so as to include the H3 Q76 side chain in its entirety.

2. *Figure 6 c: The color of the H5-NUC structure should be different than cyan for clarity since according to the authors cyan is the color used for the non-dyad linker histone (LH2) across the figures.*

The colour for the H5-NUC structure has been changed to yellow.